# MDDR: Multi-modal Dual-Attention aggregation for Depression Recognition

## ABSTRACT

Automated diagnosis of depression is crucial for early detection and timely intervention. Previous research has largely concentrated on visual information, often neglecting the value of leveraging a variety of data types. Although some studies have attempted to employ multiple modalities, they typically fall short in investigating the complex dynamics between features from various modalities over time. To address this challenge, we present an innovative Multi-modal Dual-Attention aggregation architecture for Depression Recognition (MDDR). This framework capitalizes on multi-modal pre-trained features and introduces two attention aggregation mechanisms: the Feature Alignment and Aggregation (FAA) module and the Sequence Encoding and Aggregation (SEA) module. The FAA module is designed to dynamically evaluate the relevance of multi-modal features for each instance, facilitating a dynamic integration of these features over time. Following this, the SEA module determines the importance of the amalgamated features for each frame, ensuring that aggregation is conducted based on their significance, to extract the most relevant features for accurately diagnosing depression. Moreover, we propose a unique loss calculation method specifically designed for depression assessment, named DRLoss. Our approach, evaluated on the AVEC2013 and AVEC2014 depression audiovisual datasets, achieves unparalleled performance.

## CCS CONCEPTS

• **Computing methodologies** → **Artificial intelligence**; • **Human-centered computing** → *Human-centered computing*.

## KEYWORDS

Affective Computing; Automatic depression recognition; Multimodal; Attention aggregation; Self-attention.

**ACM Reference Format:**
Anonymous Author(s). 2024. MDDR: Multi-modal Dual-Attention aggregation for Depression Recognition. In *Proceedings of Proceedings of the 32nd ACM International Conference on Multimedia (MM '24)*. ACM, New York, NY, USA, 8 pages. https://doi.org/XXXXXXX.XXXXXXX

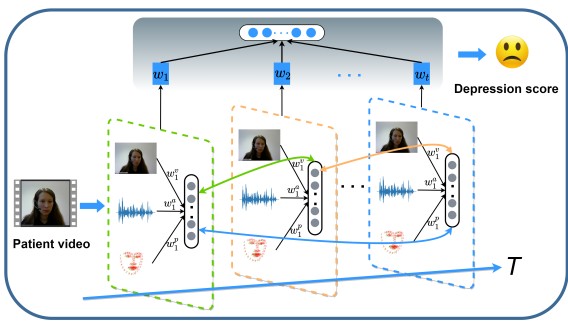

**Figure 1: Our approach employs a Multi-modal framework that effectively capture the complex dynamics between features from various modalities over time, aimed at achieving accurate and efficient automatic depression assessment.**

## 1 INTRODUCTION

Major Depression Disorder (MDD) is a severe mental illness, with over 350 million people worldwide suffering from it [1]. However, due to factors such as social stigma and a significant imbalance between the number of doctors and patients, over 50 percent of individuals with depression do not receive relevant mental health services, missing the optimal time for intervention [2, 3]. In fact, diagnosing depression is a highly complex and time-consuming task.Clinicians must comprehensively consider multiple diagnostic outcomes, including patient interviews and depression test scales, to assess the symptoms and severity of a patient's depression [4]. Nonetheless, such assessments, which heavily rely on the subjective analysis of clinicians, are prone to errors and inefficiency.

In recent years, significant advancements have been made in the research on automatic depression diagnosis based on patient facial videos, thanks to the rapid development of computer vision and deep learning technologies. Initial research primarily focused on analyzing video data of patients, utilizing basic models such as convolutional neural networks (CNNs) to detect facial features indicative of depression [7, 8, 17, 21, 23]. However, subsequent findings have revealed that the indicators of depression are not confined to visual cues alone but also include the patient's speech patterns and head movements [5, 6, 45, 46]. These elements are vital as they provide significant insights into an individual's emotional and cognitive states, thereby presenting a more comprehensive set of indicators for depression.

Recent efforts have sought to incorporate multiple modalities, analyzing features within individual modalities before integrating them [16, 18, 26, 27, 42]. This approach, however, tends to overlook the nuanced diagnostics practiced in clinical settings, where clinicians assess patients' conditions through a combination of verbal and non-verbal cues and dynamically adjust their focus based on the context. This situation highlights the need for methodologies that not only merge multimodal features but also adaptively prioritize their importance over time, mirroring the variable emphasis clinicians place on different aspects of patient behavior. To date,

 Anon.

existing research has yet to fully embrace this approach, marking a significant gap in the development of depression recognition systems that effectively replicate clinical diagnostic processes.

In this paper, we aim to bridge the identified gap by proposing a model that dynamically weights multimodal features, offering a more nuanced and clinically aligned method for depression recognition, as illustrated in Figure 1. We introduce a novel Multi-modal Dual-Attention Aggregation framework for this purpose, referred to as MDDR. To enhance the generalizability of our model, we have employed pre-trained models to extract features related to facial expressions, acoustic properties, and head movements separately.

To explore the complex dynamics between features from various modalities over time, we introduce two attention-based modules: the Feature Alignment and Aggregation (FAA) module and the Sequence Encoding and Aggregation (SEA) module. The FAA module utilizes an attention mechanism to adaptively determine the importance of each modality at each timestep, leveraging these weights to compile a fused multimodal representation for each moment. The SEA module employs a self-attention mechanism to identify interrelations among fused features at different moments, generating an aggregated representation that reflects the significance of these features throughout the entire sequence.

Furthermore, we recognize that the actual diagnosis of depression more accurately resembles an interval prediction challenge. To address this, we propose a novel loss function specifically designed for depression recognition, termed the Depression Recognition (DR) loss. The DR loss introduces a range discrimination hyperparameter, aiming to diminish the model's sensitivity to outliers and ensure the convergence of prediction results within a specified range.

Our contributions can be summarized as follows:

- We propose a novel Multi-modal Dual-Attention Aggregation architecture for Depression Recognition (MDDR), offering a more nuanced and clinically aligned approach to diagnosing depression.
- We propose two Attention Aggregation modules (FAA and SEA), designed for the dynamic fusion of features from various modalities over time.
- We have introduced a novel loss function specifically designed for depression recognition, named Depression Recognition (DR) loss.
- We evaluated our approach on two depression audiovisual datasets: AVEC2013 and AVEC2014, both achieving state-of-the-art results.

The rest of this paper is organized as follows. In Section 2, we briefly summarizes the related works for depression. In Section 3, we detail our proposed method (MDDR). Section 4 and Section 5 illustrate the experimental setup, along with the evaluate and analyze results, respectively. Finally, in Section 6, we discuss the conclusions of this paper and future work.

## 2 RELATED WORK

Considering the challenges faced in existing research on automated depression diagnosis using patient facial videos, we can categorize them into the following types.

**Multi-modal learning.** Recently, Multi-modal methods have made valuable progress in many depression recognition tasks [12,

13, 22]. For instance, Zhang et al. utilized state-of-the-art models to extract Multi-modal (audio and visual) features, then fused these features using a Transformer Encoder, achieving excellent results in multiple emotion analysis tasks [14]. Sun et al. used multilayer perceptrons (MLPs) instead of self-attention to mix across three dimensions (modality, sequence length, features) for tasks like sentiment analysis and depression estimation. [15]. Chao et al. concatenated audio features, facial appearance features, and facial shape features for early exploration of multimodal learning in depression recognition [18]. Chen et al. employed audio, textual, and visual features to design a feature fusion framework both within and across modalities, achieving advanced results in depression recognition [16]. However, the complex relationships between modalities over time have not been effectively explored. In this paper, we designed two Attention Aggregation modules to dynamically fuse features from various modalities over time, achieving a more refined multimodal integration.

**Spatio-temporal modeling.** To fully utilize the spatio-temporal information in patients' videos, researchers employ Dual-stream approaches or recurrent neural networks (RNN, LSTM) to model the spatio-temporal representation of the entire video [18–20]. For instance, Azher et al. utilized Bi-directional Long Short-Term Memory networks (BiLSTM) to capture the global representation of entire videos for depression recognition [20]. Jazaery et al. employed RNNs to model the local and global spatio-temporal information of continuous facial expressions to identify patients' depressive states [7]. Zhu et al. and Melo et al. propose a depression assessment method based on a two-stream framework, where one stream learns static features in the video through convolutional neural networks, and another stream learns dynamic features, combining results from both streams to diagnose depression [19, 29]. However, these methods focusing solely on the variation of spatial features temporally, overlooking the interrelations between spatial features at different times, represent a significant oversight. In this paper, we employ a self-attention mechanism to capture the interrelations among spatial features at different times, obtaining an enhanced representation that incorporates global relationships and spatio-temporal information.

**Pre-trained models.** Due to the scarcity of publicly available depression video datasets, researchers utilize pre-trained models to extract facial depression features from videos [19–21]. For instance, Azher et al. used the Inception-ResNet-v2 network, pretrained on the ImageNet dataset, to extract static features from videos for depression assessment [20]. Melo et al. employed a two-stream framework, with both streams utilizing the ResNet-50 network, pretrained on the VGG Face dataset, to capture appearance and motion information from videos for depression assessment [19]. Zhou et al. utilized a ResNet network trained on the CASIA large-scale facial database, subsequently fine-tuning it to capture facial depression features in patients [17]. Jazaery et al. and Melo et al. pre-trained the C3D network on the Sports-1M and UCF101 video datasets to learn depression representations in videos [7, 21]. However, these methods use pret-rained data that significantly differ from the depression recognition task, limiting the improvement in model performance. Considering the strong correlation between facial emotional states and depression outcomes, in this paper, we first utilize a pre-trained model, trained on a large-scale facial emotion

dataset, to extract facial depression features from videos, thereby enhancing the model's generalization ability.

## 3 METHODOLOGY

### 3.1 Overview

Our framework takes a face video containing audio as input. As illustrated in Figure 2, the Multi-modal features pre-extraction module extracts three modal features from given video: visual features $X_v$, acoustic features $X_a$ and Head movement features $X_p$. The Features Alignment and Aggregation (FAA) module aligns and stacks these features into $X \in \mathbb{R}^{L \times M \times D}$, where $L$ represents the length of the sequence, $M$ represents the number of modalities, and $D$ represents the feature dimension. The $X$ is processed by the Inter-modality Attention aggregation network to produce $F \in \mathbb{R}^{L \times D}$. Following this, the Sequence Encoding and Aggregation (SEA) module uses a self-attention-based Encoder to process $F$ and produce $O \in \mathbb{R}^{L \times D}$, followed by the Temporal Attention aggregation network that yields an aggregated representation $E \in \mathbb{R}^D$. Finally, $E$ is processed by a classification network based on a Multilayer Perceptrons (MLPs) to produce the final depression assessment results, which is a depression score ranging from 0 to 63.

### 3.2 Multi-modal features pre-extraction

To derive more generalized features indicative of depression, we utilize pre-trained models to extract deep representations from each modality within raw audiovisual data. This paper focuses on extracting three types of features: visual, acoustic, and head movement features, each demonstrating unique characteristics associated with depression. For the extraction of head movement and acoustic features, we employ OpenFace[34] and HuBERT[35], respectively. Specifically, considering the strong correlation between depression-related visual features and facial expressions, coupled with the challenge conventional image pre-trained models (such as those based on ImageNet[44]) face in extracting effective emotional features, we have developed a facial feature extraction model. This model, which is pre-trained using ResNet-50[47] on the extensive emotion recognition dataset AffectNet[43], achieves a generalized deep representation of facial expressions.

### 3.3 Features alignment and aggregation (FAA)

Due to variations in the methods used for pre-extracting features across different modalities, there can be discrepancies in sequence lengths and feature dimensions. These differences pose challenges for the subsequent fusion of information among modalities. To align the acoustic and head movement features with the visual features, we utilize two alignment networks that process these two types of features separately. Both alignment networks employ a 1D-CNN as their backbone, enabling them to effectively capture the local features within the input sequence. After processing, we obtained three modalities of features with aligned dimensions, including visual features $X_v \in \mathbb{R}^{L \times D}$, acoustic features $X_a \in \mathbb{R}^{L \times D}$ and head movement features $X_p \in \mathbb{R}^{L \times D}$.

In diagnosing depression, clinicians dynamically focus on patients' depressive characteristics at different modalities over time and considering them comprehensively based on the context. To emulate this complex diagnostic approach, we utilize an attention mechanism to adaptively determine the importance of each modality at each timestep, leveraging these weights to compile a fused multimodal representation for each moment. Referred to as Inter-modality Attention aggregation network, as illustrated in Figure 3 (a). First, we stack these aligned features from different modalities, denoted as $X \in \mathbb{R}^{L \times M \times D}$. This step is primarily for the ease of subsequent unified data processing, with the attention weights for each modality learned separately. Next, we utilize a multi-layer fully connected network with shared weights to learn the attention weights of each modality at different times, which are then normalized using a softmax function. Finally, we multiply the attention weights of each modality by their respective feature vectors and sum these products, obtaining a fused feature that incorporates the significance of each modality over all times, defined as $F \in \mathbb{R}^{L \times D}$. This method involves a linear combination of each feature based on attention weights, proves to be much faster and more space-efficient in practice. The Attention aggregation calculation formula is as shown in Equation 1, where $W_m$ represents a trainable parameter, $W_m \in \mathbb{R}^{L \times 1 \times D}$.

$$F = \sum_{i=0}^{M} softmax(W_m X^T) X \tag{1}$$

### 3.4 Sequence encoding and aggregation (SEA)

The complex interrelations between depressive characteristics at different times may contain subtle clues of depression. To capture these clues, we utilize a self-attention-based Encoder module to encode the input sequence. The Encoder consists of multiple Blocks, each comprised of a self-attention network and a fully connected layer. Within a Block, the process begins with a Multi-Head Attention layer, followed by a residual connection and Layer Normalization, then fed into a Feed Forward network, with multiple Blocks repeating this process to finally output a sequence. Additionally, the input vectors are not fed directly into the Encoder but are first augmented with Positional Encoding, which is crucial for processing time-series data. The self-attention calculation formula is as shown in Equation 2, where $Q$, $K$, and $V$ represent the query, key, and value vectors, respectively. $O \in \mathbb{R}^{L \times D}$ denotes the output obtained from self-attention computations that encompasses global relationship information.

$$O = softmax(\frac{QK^T}{\sqrt{d_k}})V \tag{2}$$

We implemented a method similar to the Inter-modality attention aggregation described in Section 3.3 to aggregate temporal information, referred to as Temporal attention Aggregation network, as illustrated in Figure 3 (b). The Temporal Attention aggregation network dynamically focuses on the significance of each moment within the input sequence, condensing long sequences into a fixed representation that encompasses the significance of all temporal aspects. $O$ , after the Temporal Attention Aggregation network, is transformed into a fixed-size depression representation $E \in \mathbb{R}^D$. Finally, this representation is passed through a classification network based on Multilayer Perceptrons (MLPs) to output the final depression assessment results.

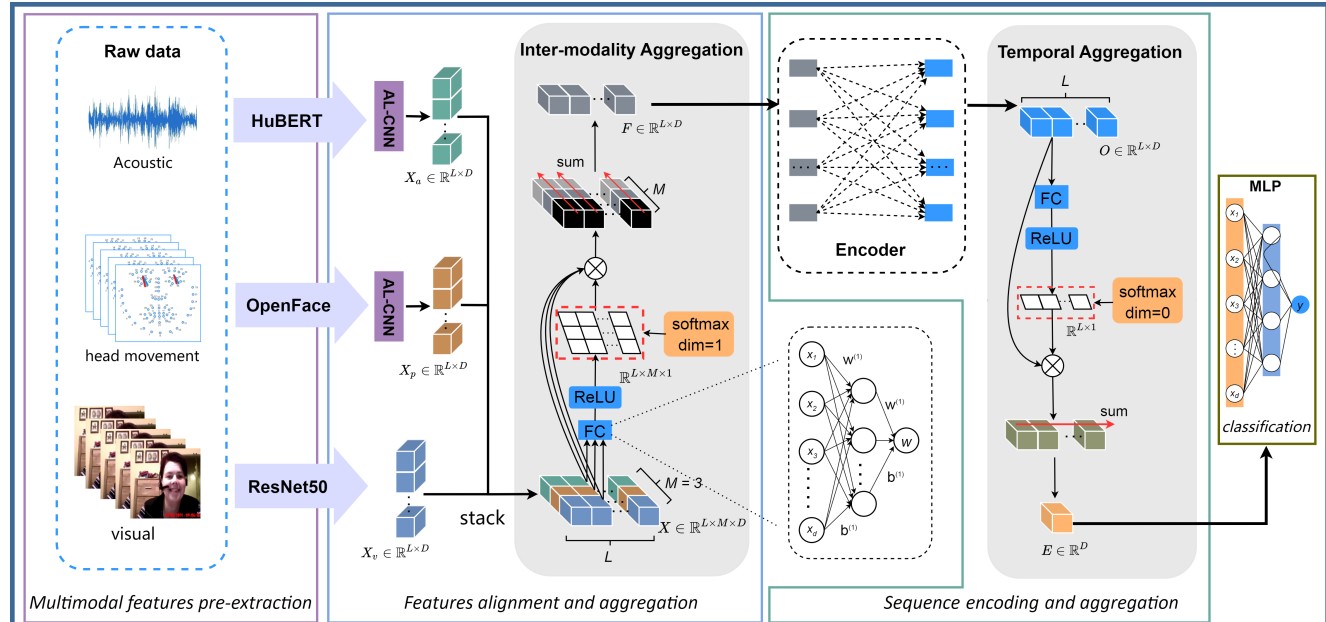

**Figure 2: Overall framework of the proposed method MDDR. The Multi-modal feature Extraction module extracts multiple pre-defined features from raw audiovisual data, including visual features $X_v$, acoustic features $X_a$ and head movement features $X_p$. The FAA module aligns and stacks these features into $X \in \mathbb{R}^{L \times M \times D}$. Processed by the Inter-modality Attention aggregation network to produce $F \in \mathbb{R}^{L \times D}$. After that, the SEA module uses a self-attention-based Encoder to process $F$ and produce $O \in \mathbb{R}^{L \times D}$, followed by the Temporal Attention aggregation network that yields a representation $E \in \mathbb{R}^D$. Finally, $E$ is processed by a classification network to produce the final depression assessment results.**

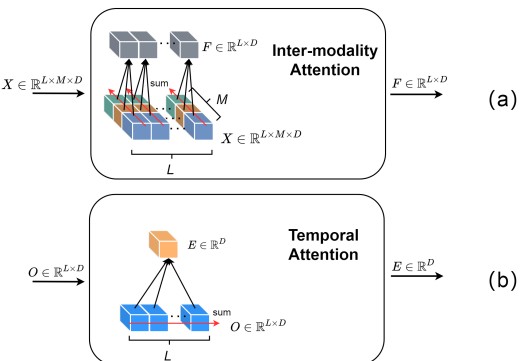

**Figure 3: We utilized two different Attention aggregation network to separately aggregate information across modalities and over temporal sequences. (a) The Inter-modality Attention aggregation Network. (b) The Temporal Attention aggregation network.**

## 3.5 Depression Recognition Loss (DR)

In clinical practice, depression scores obtained through interviews and scales are usually range values, and doctors categorize scores within the same range into one category [39]. Therefore, we designed a loss calculation method suitable for depression assessment (DR), which introduces a range discrimination hyperparameter $\delta$ aimed at reducing the model's sensitivity to outliers and ensuring convergence of prediction results within a specified range. Specifically, when the predicted value and the target value are in the same category and the prediction error is less than $\delta$ ($loss \leq \delta$), the model's predictions can be considered consistent with clinical diagnoses. In this case, we multiply the loss by a very small weight $w$ to minimize the gradient impact from the current sample. In all other cases, the Huber loss is applied to calculate the loss. Huber loss is a loss calculation method that combines the advantages of Mean Square Error (MSE) and Mean Absolute Error (MAE), offering robustness against outliers.[40].

$$L_\delta(y, f(x)) = \begin{cases} wL_H, & \text{if} (y, f(x)) \in C \wedge |y - f(x)| \leq \delta \\ L_H, & \text{other} \end{cases} \quad (3)$$

The DR loss calculation formula as shown in Equation 3 , where $y$ represents the target value, $f(x)$ represents the predicted value. $C$ denotes a specific depression category, $L_H$ is the Huber loss function. $w$ is a minimal weight value, set here as 0.1. The hyperparameter $\delta$ can be selected through cross-validation for optimal value.

# 4 EXPERIMENTAL SETUP

## 4.1 Dataset

We conducted extensive experiments on the AVEC2013 and AVEC2014 audiovisual depression datasets [30, 31]. These are the currently known two public datasets containing original video data of patients with depression, providing conditions for our research to capture depressive features from raw audiovisual data. The AVEC2013 depression dataset comprises 150 videos from 82 participants, with an average video length of about 25 minutes. Each video documents participants performing a series of tasks, including sustained vowel phonation, speaking loudly, counting from 1 to 10, among others. The AVEC 2014 depression dataset contains 300 recorded videos from 84 participants, with an average video length of about 2 minutes. It encompasses two types of recordings: one in which participants loudly read excerpts from the German fable "The North Wind and the Sun" and another where they answer routine questions.

Each video is labeled with an assessment based on the Beck Depression Inventory (BDI-II), representing its level of depression. The BDI-II is the most widely used self-rating questionnaire related to depression, extensively employed for detecting depressive symptoms in the general population and assessing the severity of depression in psychiatric patients [38]. It consists of 21 questions measuring aspects of depression, including typical symptoms such as sadness, guilt, suicidal thoughts, and lack of interest. BDI-II scores range from minimal (0-13), to mild (14-19), moderate (20-28), and severe (29-63).

## 4.2 Evaluation Metrics

The model performance is evaluated using Mean Absolute Error (MAE) and Root Mean Square Error (RMSE), both of which are used to measure the degree of deviation between predicted values and target values. RMSE represents the square root of the average squared differences between predicted values and target values. MAE represents the average absolute deviation between predicted values and target values. The definitions of MAE and RMSE are as follows:

$$\text{MAE} = \frac{1}{n} \sum_{i=1}^{n} |x_i - \hat{x}_i| \qquad (4)$$

$$\text{RMSE} = \sqrt{\frac{1}{n} \sum_{i=1}^{n} (x_i - \hat{x}_i)^2} \qquad (5)$$

Where $N$ is the total number of samples, $\hat{x}_i$ is the predicted value for the $j$-th sample, and $x_i$ is the true value for the $i$-th sample.

## 4.3 Implementation details

Considering the redundancy in video data, we extracted one frame out of every 100 for AVEC 2013 and four frames out of every 100 for AVEC 2014, cropping the images to a size of 224×224 pixels. We utilize facial emotion pre-trained model to extract visual features from facial images, resulting in 2048-dimensional deep representations. We use the audio processing library librosa to extract audio signals from videos with a sampling rate of 16000 Hz. We utilize speech pre-trained model to extract acoustic features, resulting in 1024-dimensional deep representation. Using the open-source tool

OpenFace, we extract head pose and gaze features from the face images after frame extraction, obtaining a feature dimension of 12. For missing sequence features, we use zero-padding to ensure that each sample sequence length is consistent.

We employed the PyTorch deep learning framework for all experiments, which were conducted on AMD Ryzen 9 5900HX with GeForce RTX 3080 hardware. In our method, we adopted the Stochastic Gradient Descent (SGD) algorithm with a momentum of 0.9 and a weight decay of 0.0005. The initial learning rate was set to 0.001, employing learning rate decay (the learning rate is multiplied by 0.1 every 50 training rounds). The batch size was set to 16. We utilized DRLoss as the loss function, with hyperparameters set to 2. Data normalization was performed using LayerNorm, with ReLu as the activation function and a dropout rate of 0.1. The AL-CNN uses three 1D convolutional layers for acoustic features with strides (5, 5, 2), kernels (10, 5, 3), and filters (128, 256, 512), and a single layer for head movement features with a stride of 1, a kernel size of 1, and 512 filters. The hidden dimension of the sequence Encoder module is 2048, the number of heads in the multi head attention model is 8, and the number of encoder layers is 6. The classification network employs two fully connected layers, with a hidden layer dimension of 128.

## 4.4 Baseline

We introduce the baselines compared in our experiments as follows, including common base classifiers and novel deep learning methods.

**LPQ**[32, 33]: This approach employs artificial features such as Local Phase Quantization (LPQ) to extract features of patients' facial expressions. However, this method is susceptible to changes in the external environment, can only represent spatial information, and fails to capture dynamic information.

**Two-stream**[19, 29]: This approach utilizes convolutional neural networks to extract static features from video frames and employs optical flow or temporal pooling methods to capture the dynamic features of the video. Finally, the results from both streams are aggregated to determine the ultimate depression outcome.

**C3D**[7, 21, 23]: 3D convolutional neural network can simultaneously extract spatial and temporal features from 3-dimensional data, making them more suitable for video data modeling. However, the number of parameters in 3D CNN is much larger than in 2D CNN, and due to hardware, memory, and runtime limitations, they cannot model excessively long video sequences at once.

**RNN**[18, 20]: This approach uses 2D CNN to extract static features from video frames or employs 3D CNN to extract short-term spatio-temporal features from video clips. Then, recurrent neural networks (RNN, LSTM) are used to further learn global spatio-temporal information from sequential features, facilitating depression assessment.

**Attention**[17]: The authors utilize the ResNet-50 network to learn frame-level features from the video, outputting a set of feature vectors. Subsequently, two cascaded attention modules are employed to adaptively learn the weights of different facial images, generating an aggregated representation. Finally, a regression layer uses the aggregated features to produce a depression score.

**AUs**[28]: The authors use Facial Action Units (AUs) as the underlying features for each frame and employ spectral heatmaps

and spectral vectors to characterize multi-scale spatio-temporal information in videos. Subsequently, the constructed spectral representations are fed into Convolutional Neural Network (CNN) and Artificial Neural Network (ANN) for depression analysis.

## 5 EXPERIMENTAL RESULTS AND ANALYSIS

### 5.1 Comparative experimental results

We compared our proposed method with the latest results on the AVEC2013 and AVEC2014 audiovisual depression datasets, as shown in Table 1. Our method outperforms all schemes in the table, achieving state-of-the-art results. In AVEC2013, our method improved the MAE results by 3.8% compared to the previous state-of-the-art approaches. In AVEC2014, both RMSE and MAE results saw improvements of 3.5% and 0.4%, respectively, against prior state-of-the-art methods. Specifically, 1) Methods using deep neural networks surpassed those relying on manual features (LPQ) [30–33]. 2) Our approach exceeded the performance of all convolutional neural network methods [8, 21]. Such methods focusing solely on the spatial features and overlooking the temporal features, which impacts model accuracy. 3) Our method also outperformed both Dual-stream and recurrent neural network approaches [7, 20, 29]. Such methods primarily focuses on the temporal changes in facial spatial features, and the interrelations between spatial features at different times have not been adequately addressed, resulting in limited model effectiveness. 4) Our method surpassed those using other pre-trained models [17, 19, 20]. Interestingly, although both [19] and [20] employed Dual-stream approaches and used pre-trained models to extract facial features, Melo et al. utilized a face recognition pre-trained model that is more closely related to the task of depression recognition, yielding better results. Notably, we found that attention-based methods are effective [17, 25]. For example, [25] utilized two attention modules to capture facial depression characteristics in patients, achieving commendable depression detection results even without the use of pre-trained models. Furthermore, Song et al. employed additional low-dimensional features related to emotions, such as Facial Action Units (AUs) [28]. This approach also yielded favorable results, providing new insights for our subsequent research.

### 5.2 Ablation study

To further validate the effectiveness of each module we designed, we conducted extensive ablation experiments.

*5.2.1 Ablation Study on Pre-trained models.* To assess the effectiveness of facial emotion pre-trained models, we conducted experiments under three conditions: using models pre-trained on AffectNet[43], using models pre-trained on ImageNet[44], and without using any pre-trained models, as shown in Table 2. The results indicate that using facial emotion pre-trained models trained on AffectNet significantly enhances model performance, outperforming other scenarios considerably. Compared to using models pre-trained on ImageNet, RMSE and MAE improved by 14.3% and 19.2%, respectively, in AVEC2013. In AVEC2014, RMSE and MAE increased by 13.9% and 21.7%, respectively. Using facial emotion pre-trained models enhances the focus on facial expression details, thereby extracting more effective features of expressions. Such features are

| Methods | AVEC2013 | | AVEC2014 | |
|---|---|---|---|---|
| | MAE↓ | RMSE↓ | MAE↓ | RMSE↓ |
| Baseline[30, 31] | 10.88 | 13.61 | 8.86 | 10.86 |
| LPQ + Geo [32, 33] | 7.86 | 9.72 | 8.20 | 10.27 |
| Concat+LSTM[18] | \ | \ | 7.91 | 9.98 |
| Two-DCNN [29] | 9.82 | 7.58 | 7.47 | 9.55 |
| C3D+RNN [7] | 7.37 | 9.28 | 7.22 | 9.20 |
| Two-Stream[20] | 7.04 | 8.93 | 6.86 | 8.78 |
| C3D+atten[23] | 6.83 | 8.46 | 6.78 | 8.42 |
| C3D+Pool [21] | 6.40 | 8.26 | 6.59 | 8.31 |
| LGA-CNN[25] | 6.59 | 8.39 | 6.51 | 8.30 |
| ResNet + atten [17] | \ | \ | 6.37 | 8.43 |
| ResNet+distribution[8] | 6.30 | 8.25 | 6.15 | 8.23 |
| Two-stream[19] | **5.96** | 7.97 | 6.20 | 7.94 |
| AUs[28] | 6.16 | 8.10 | 5.95 | 7.15 |
| Proposed Approach | 5.98 | **7.80** | **5.75** | **7.12** |

**Table 1: Comparison of our method with recent other methods on the AVEC 2013 and AVEC 2014 datasets**

highly correlated with depression and enable the model to generalize facial depression features from limited training data, which is crucial for depression recognition tasks.

| Pre-trained | AVEC2013 | | AVEC2014 | |
|---|---|---|---|---|
| | MAE↓ | RMSE↓ | MAE↓ | RMSE↓ |
| \ | 9.86 | 10.53 | 9.45 | 10.93 |
| ImageNet | 7.13 | 8.92 | 7.0 | 8.11 |
| AffectNet | **5.98** | **7.80** | **5.75** | **7.12** |

**Table 2: Result of the study on Pre-training models**

*5.2.2 Ablation Study on multiple modalities.* To assess the impact of different modal information on depression assessment, we compared the effects of a single visual modality with those of integrated multiple modalities information, as shown in Table 3. Experiments indicate that although visual modalities have the potential to contain abundant depressive cues, indicators of depression are not confined to visual cues alone. In fact, these indicators also encompass other modalities such as the patient's speech patterns and head movements. In our experiments, integrating multimodal information, including audio and head movement, significantly enhanced model performance. These results validate our hypothesis that integrating information from various modalities enhances our comprehensive understanding of patients' mental states, thereby improving the accuracy of depression detection.

*5.2.3 Ablation Study on Dual-Attention aggregation module.* To assess the effectiveness of our proposed Dual-Attention aggregation modules, we conducted experiments on depression assessment results with the FAA module removed, the SEA module removed, and both modules in use, as shown in Table 4. The results show a significant decline in performance when either module is removed. This

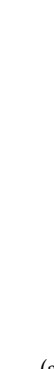
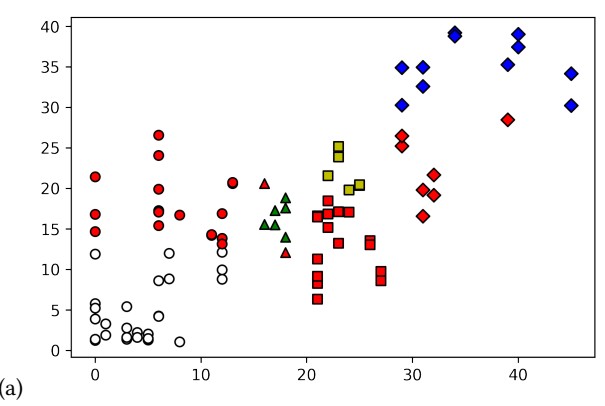
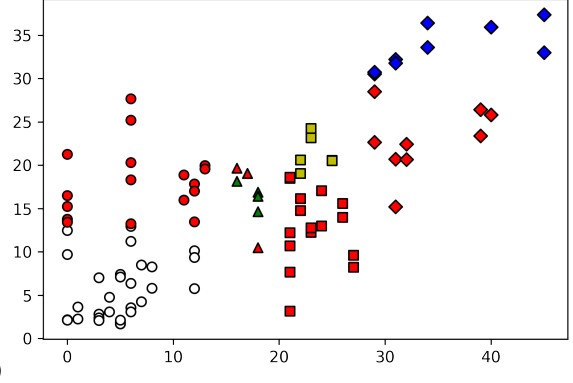

(a) (b)

**Figure 4: The experiment was conducted on the AVEC2014 dataset, employing different loss functions to obtain depression assessment results. Different colors represent different categories of depression severity, with red indicating outliers where the predicted values do not match the target value category. (a) represents the depression assessment results using DR loss. (b) represents the depression assessment results using Huber loss.**

| Model | AVEC2013 | | AVEC2014 | |
|---|---|---|---|---|
| | MAE↓ | RMSE↓ | MAE↓ | RMSE↓ |
| $v$ | 6.32 | 8.11 | 6.21 | 7.76 |
| $a$ | 8.55 | 10.25 | 8.76 | 9.94 |
| $p$ | 9.53 | 10.24 | 9.0 | 10.01 |
| $v+a+p$ | **5.98** | **7.80** | **5.75** | **7.12** |

**Table 3: Result of the study on multiple modalities**

confirms our hypothesis: Firstly, the characteristics of depression in patients change over time across various modalities in depression diagnosis. The FAA module dynamically learns the importance of different modalities over time, enabling effective integration of multimodal information. Secondly, patients' depressive manifestations are often highly concealed. The SEA module, by capturing the interrelations among features fused at different times within videos, enabling the detection of subtle depression clues hidden within complex temporal relationships. These considerations are crucial for a detailed and accurate depression diagnosis. Notably, the method integrating both FAA and SEA modules achieved the best results, with RMSE and MAE reaching 7.12 and 5.75, respectively.

| Methods | AVEC2013 | | AVEC2014 | |
|---|---|---|---|---|
| | MAE↓ | RMSE↓ | MAE↓ | RMSE |
| FAN | 6.85 | 8.55 | 6.22 | 7.65 |
| SAN | 6.68 | 8.36 | 6.13 | 7.32 |
| FAN+SAN | **5.98** | **7.80** | **5.75** | **7.12** |

**Table 4: Result of the study on the Dual-Attention aggregation module**

*5.2.4 Ablation Study on DR Loss.* To verify the effectiveness of our proposed DR loss, experiments were conducted to compare

depression assessment results using different loss calculation methods. According to the classification method in Section 4.1 using the BDI-II, depression severity is categorized into four levels: minimal (white, circle), mild (green, triangle), moderate (yellow, square), and severe (blue, diamond) [38]. Red dots represent outliers, indicating data points that do not fall into the same category as their actual depression level. All true values and predicted values were plotted on one graph, as illustrated in Figure 4. The results show that the number of outliers with the Huber loss method was 47, while with the DR loss method, it was 43. Our loss calculation method resulted in fewer outliers and depression scores within the same category being closer together, which also aligns more closely with clinical diagnostic outcomes in real-world scenarios.

## 6 CONCLUSION

In this paper, We propose a novel Multi-modal Dual-Attention aggregation architecture for depression recognition (MDDR). It extracts multiple modalities features from patient video as inputs and employs two Attention aggregation modules to capture the complex dynamics between features from various modalities over time, facilitating accurate and efficient automatic depression assessment. Additionally, we propose a loss calculation method (DR) suitable for depression assessment. Our method was evaluated on two standard depression audiovisual datasets (AVEC2013 and AVEC2014), demonstrating its effectiveness and superiority.

Due to the scarcity of data, the method of extracting depression features from video data using complex deep neural networks is limited. In future work, we will focus on mining more effective and generalized depression features from small sample data. Furthermore, we will collect more data from depression patients and establish a multimodal database of depression patients (including EEG, physiological indicators, etc.) to enhance our research.

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
