# OpenReview forum: "MDDR:Multi-modal Dual-Attention aggregation for Depression Recognition"
_acmmm.org/ACMMM/2024/Conference — MM2024 Oral_

### Official Review · Reviewer_qnr7 · 2024-05-07

**Rating:** 4
**Confidence:** 3

**Summary:**

This paper designed two attention aggregation modules to consider the multi-modal feature and the temporal dynamics for depression detection.  The study presents a clear and deep background literature and the methodology proposed is clearly stated.

**Strengths:**

1. This paper depends on pre-trained models to extract features that overcome the overfitting problem.
2. An FAA module is designed to align the multi-modal feature and an SEA module is designed to capture the temporal dynamics.
3. The paper is well-written and presented clearly. The proposed method can be more accurate in two datasets.

**Limitations:**

1. More details about the pre-trained model used for uni-modal feature extraction can be provided.
2. How to use 1-D CNN for feature alignment if there are discrepancies in sequence lengths?
3. Why the attention used in the FAA and in the SEA are two different types? Will it be better to use the same attention mechanism or worse?
4. There may be a mistake in Table 4 about the name of the methods.
5. The experimental results show that the proposed DR loss can only bring 4 fewer outliers. What about the MAE and RMSE? In addition, the effect of the hyper-parameter $\delta$ and $w$ in the DR loss should be verified.

**Suitability:**

3

---

### Official Review · Reviewer_4TT1 · 2024-05-21

**Rating:** 5
**Confidence:** 3

**Summary:**

Objective: In this paper, the authors propose an innovative multimodal dual aggregation network for depression detection. The authors make use of multimodal pretraining features and propose two attention networks: the feature alignment and aggregation (FAA) module and a Sequence Encoding and Aggregation (SEA) module. Both of these modules are designed to evaluate the relevance of multimodal features for each instance facilitating dynamic integration of these features and determining the significance of each of these features. The authors also propose DR Loss suitable for depression assessment and evaluate their proposed methodology on AVEC 2013 and AVEC 2014 datsets.

The authors use three different kinds of features – visual, acoustic, and head movement features extracted from pre-trained convolutional models, OpenFace and HuBERT respectively.

**Strengths:**

The paper is well written and the architecture is neatly explained.

Main contributions: dual attention aggregation network for alignment of various multimodal features, and determine weights for the appropriate fusion of modalities; propose the use of DR loss.

**Limitations:**

1.	Over a decade, OpenFace was used to extract visual features. And they were found to be robust for the task. It is not clear why pre trained models based on conv2D were employed specifically to derive visual features.
2.	Head movement features – head movement is also a visual feature. Why there is a separate stream for head movement? Other visual features such as gaze are not included. If separate attention networks are considered, architecture becomes very complicated.
3.	Authors use a simple method to handle missing modality. Better strategies could have been employed.
4.	Why textual features are not considered? In AVEC 2014, participants answer casual questions. According to [1], [2], [3] and many previous methods, textual features proved to be the most significant modality among others.
5.	Two datasets considered here are very different. The average duration for AVEC 2013 is 23 min while for AVEC 2014 is 2min. Is there any difference in the performance because multimodal alignment is challenging in AVEC 2013.
6.	Evaluation of the proposed architecture could have been performed on multiple feature sets.
References:
1.	Multimodal depression detection using task-oriented transformer-based embedding
2.	Multi-level attention network using text, audio and video for depression prediction
3.	The Verbal and Non-Verbal Signals of Depression–Combining Acoustics, Text and Visuals for Estimating Depression Level

**Suitability:**

2

---

### Official Review · Reviewer_m7VU · 2024-05-24

**Rating:** 2
**Confidence:** 2

**Summary:**

The paper presents a novel architecture called Multi-modal Dual-Attention Aggregation for Depression Recognition (MDDR), designed to automatically diagnose depression by analyzing multimodal data such as audio and visual cues from patients. The MDDR framework utilizes pre-trained models to extract features related to facial expressions, acoustic properties, and head movements. It introduces two attention aggregation mechanisms: (1) Feature Alignment and Aggregation (FAA) Module: This module dynamically evaluates the relevance of multi-modal features for each instance, facilitating the dynamic integration of these features over time. (2) Sequence Encoding and Aggregation (SEA) Module: This module determines the importance of the combined features for each frame, ensuring that the aggregation is conducted based on their significance to extract the most relevant features for depression diagnosis. Additionally, the paper proposes a unique loss calculation method called Depression Recognition Loss (DRLoss), which is designed specifically for depression assessment.

**Strengths:**

The MDDR framework introduces a novel approach to depression recognition by utilizing a dual-attention mechanism that dynamically evaluates and integrates multimodal features over time. This work describes a technically sound methodology, with clear explanations of the pre-trained models used for feature extraction, the alignment and aggregation of multimodal features, and the sequence encoding process. The introduction of the Depression Recognition Loss (DRLoss) is also a technically appropriate solution to address the specific challenges of depression assessment.

**Limitations:**

1. In the experimental design, the baseline comparison object selected by the authors is missing some SOTA work. such as: \
(1) de Melo W C, Granger E, Lopez M B. MDN: A deep maximization-differentiation network for spatio-temporal depression detection [J]. IEEE Transactions on Affective Computing, 2021, 14(1): 578-590. \
(2) Niu M, Tao J, Liu B, et al. Multimodal spatiotemporal representation for automatic depression level detection [J] IEEE Transactions on Affective Computing, 2020, 14(1): 294-307.\
(3) De Melo W. C., Granger E., and Hadid A. A deep multiscale spatiotemporal network for assessing depression from facial dynamics [J]. IEEE Transactions on Affective Computing, 2020, 13(3): 1581-1592.\
Based on the results of these references, this work cannot be concluded to achieve the level of sota.
2. In the ablation experiments, the authors only set the comparison using single-modal features and three-modal features. The case of any two combinations should also be taken into account.
3. FAN and SAN in Table 4 appear for the first time and should be explained with abbreviations. Or does the author mean FAA and SEA?
4. The authors compared the effect of extracting facial expressions from models pre-trained on different datasets on the final results but did not conduct similar experiments for the other two modalities. I think authors should consider comparing the impact of different feature extraction networks on the final results of this framework.

**Suitability:**

3

---

### Meta-Review · Area_Chair_S9TE · 2024-07-10

**Recommendation:** Accept (Oral)
**Confidence:** 5

**Metareview:**

this paper investigates the problem of depression recognition and introduces a new approach called Multi-modal Dual-Attention Aggregation for Depression Recognition (MDDR). to consider the facial expressions, acoustic properties, and head movements jointly, this paper introduces two attention mechanisms, and achieves competitive performance.

initially, the paper received ratings of WR, WA, and BA. the reviewers commended the paper for the dual attention module, the DR loss, and the overall writing. they also voiced their concerns over the missing comparison against SOTAs, ablation and variant study, clarity issues, and need for discussion on topics including visual features and the two types of attentions. during rebuttal, the authors mostly addressed these concerns and the final ratings improved to WA, WA, and WA.

since the AC was not able to finish the meta-review in time, the PC stepped in and went through all the reviews and rebuttal. after careful consideration, the PC recommends Accept.